

# Nighttime migrations and behavioral patterns of *Pempheris schwenkii*

Keita Koeda[1,2], Hideyuki Touma[3] and Katsunori Tachihara[4]

[1] The University Museum, The University of Tokyo, Tokyo, Japan
[2] Previously: Graduate School of Engineering and Science, University of the Ryulyus, Nishihara, Okinawa, Japan
[3] Okinawa Churaumi Aquarium, Motobu, Okinawa, Japan
[4] Faculty of Science, University of the Ryukyus, Nishihara, Okinawa, Japan

Corresponding author
Keita Koeda,
koeda@um.u-tokyo.ac.jp

## ABSTRACT

**Background:** Although the biomass of the nocturnal fishes is almost same as that of diurnal fishes, most of the ecological studies that examine feeding or reproductive behaviors are on diurnal fishes. Therefore, there is limited ecological information regarding the nocturnal fishes. This fact may be attributed to the difficulty in observing them during darkness. Members of the genus *Pempheris* (Pempheridae) are one of the most abundant nocturnal fishes on coral reefs.

**Methods:** The nighttime migrations of *Pempheris schwenkii* were observed by attaching a chemical luminescent tag. Tagged fishes were followed by an observer without torch and SCUBA, and their positions and estimated depths were plotted on an underwater topographic map. Aquarium tank observation was carried out to further describe their habits during the night.

**Results:** The new tagging method provided good data for observing the migration behavior. In all five observations, the target fishes started nighttime migration from the entrance of their cave within 1 h after sunset. All of them immediately left the inner reef and spent most of the observation time near the surface (0–5 m depth) or shallow (5–15 m depth) water-columns of the outer reef. Their migration pattern varied between days, but they migrated long distance (379–786 m/h) during each observation. The behavior observed in the aquarium tank was categorized into five patterns: schooling, shaking, migrating, spawning, and feeding. Shaking and spawning were observed during one of three observation days.

**Discussion:** The present study firstly clarified the small-scale but dynamic nocturnal migration pattern of *P. schwenkii* in nature by a new method using chemical luminescent tags. In addition, combined observations from nature and an aquarium could be used to estimate the behavior of this species. *Pempheris schwenkii* may reduce their predation risk of eggs and adults by spawning at outer reef in nighttime. It was estimated that they can potentially migrate 4–7 km/night. The rapid growth known for this species may have been supported by their feeding behavior where they can fill up their stomach every night with rich zooplankton in outer reefs. Furthermore, the behavior of this species indicates the possibility that they make an important contribution to the flow of energy and materials in their coral reef ecosystem.

**How to cite this article** Koeda K, Touma H, Tachihara K. 2021. Nighttime migrations and behavioral patterns of *Pempheris schwenkii*.
PeerJ 9:e12412 DOI 10.7717/peerj.12412

## INTRODUCTION

In tropical and subtropical inshore waters, the 24-h day is partitioned by two largely different groups of fishes, diurnal fishes and nocturnal fishes. In general, both groups hide in their respective shelters during their inactive periods. *Holzman et al. (2007)* showed the importance of nocturnal zooplanktivorous fish that connect lower order and higher order consumers on the food chain. In addition, they indicated that the biomass of the nocturnal fishes is almost the same as that of the diurnal fishes. However, most of the ecological studies that are focused on feeding, trophic or reproductive behaviors were conducted on diurnal fishes, and ecological information regarding nocturnal fishes is relatively limited due to the difficulty of observing them in the nighttime (*e.g.*, *Bray, Miller & Geesey, 1981*). Very few studies have been done on the nighttime behavior of nocturnal fishes (*e.g.*, *Hobson, 1972*), and most of those are mainly focused on diurnal fishes.

Fishes of the genus *Pempheris*, sweepers, comprise one of the most abundant nocturnal groups of fish in rocky and coral reefs. Characteristically, the fishes of this genus hide in underwater caves, rock recesses, or crevices during the day, and swim out to open water at night, where they primarily prey on zooplankton (*Fishelson, Popper & Gunderman, 1971*; *Gladfelter, 1979*; *Golani & Diamant, 1991*; *Fishelson & Sharon, 1997*; *Platell & Potter, 1999*, *2001*; *Annese & Kingsford, 2005*; *Sazima et al., 2005*). *Fishelson, Popper & Gunderman (1971)* firstly tried to determine the leaving and the arriving movements of sweepers in the Red Sea and observed their behavior near their daily shelters around sunset and sunrise. *Annese & Kingsford (2005)* compared the migrating systems and feeding habits of *P. affinis* and *P. multiradiata* in Australia on the basis of observation around the diurnal site combined with the analysis of the stomach contents of collected specimens. *Fishelson & Sharon (1997)* observed the migration of juvenile sweepers in the Red Sea. The difficulty in the observation of nocturnal fishes may be attributed mainly to their behavior such that they are repelled by an observer's torch-light. Therefore, this warrants the need of nighttime underwater observation without light to observe the unaffected behavior of the nocturnal fishes. In another study, *Gladfelter (1979)* tried to observe the nighttime migrations of *Pempheris schomburgkii* in the Atlantic Ocean on the basis of direct observations under moonlight. However, this method was only suited for use in the shallow reef area, slow speed movement, and bright moon, making it difficult to reduce the interference from the observer, because the observer needs to swim close to the target fish. In the present study, we used chemical luminescent tags to observe sweepers under the darkness in natural conditions. With this method, we could observe the migrations of tagged fish without any apparent effects on their behavior at depths up to 20 m. Although the method can trace the migration of nocturnal fishes, it was impossible to observe their extensive behavior such as feeding or reproduction. Therefore, these behaviors were observed in a large aquarium, and the data was combined with the natural migration pattern to estimate the sequence of dynamic nighttime migration of *Pempheris schwenkii*,

which is one of the most common species in the West Pacific, to elucidate their reproduction and feeding patterns, and to evaluate their potential roles in the coral reef ecosystem.

## MATERIALS & METHODS

### Study area

The nighttime migrations of *P. schwenkii* were observed in and around the half-underwater cave nearby Cape Maeda (26°26′N, 127°46′E) on the western coast of Okinawa-jima Island in the Ryukyu Archipelago. The length, width, and depth of the huge half-underwater cave were up to ca. 50 m, 10 m, and 5 m, respectively. Sunlight comes in from outside at the entrance of the cave, and the water and the walls in the cave reflect a blue light. This cave is called "Ao-no-dokutsu", which means "blue cave" in Japanese, and is one of the most popular diving spots in Okinawa. It contains a number of rock recesses, and several hundred to more than a thousand (based on the seasons) of *P. schwenkii* inhabit this cave. Approximately 50 individuals of *P. adusta* and one or two individuals of *P. oualensis* were observed near the entrance of the cave.

A general underwater topographic map of the study area was prepared by combining depth data from a depth finder (Mistral Instruments) and lat/long data from a GPS (Garmin eTrex Venture HC). Directions in the present study were shown from 0–360°, with 0° and 180° meaning north and south, respectively. The coastline of the study area ran straight from northwest (310°) to southeast (130°). The site contains a narrow inner reef (<5 m depth) and grows rapidly deeper (up to 20 m) at the outer reef. The border between inner and outer reefs ran parallel 40–50 m from the coastline. The depth was slightly deeper to the offshore in outer reef by up to 30–40 m.

Time of sunset for each observation day was obtained from the website of Ephemeris Computation Office (https://eco.mtk.nao.ac.jp/koyomi/index.html.en).

### Field observations

The migration and behavioral patterns of *P. schwenkii* were observed for a single fish each day for 5 days during 22nd May 2013 (11.6 age of moon) to 8th June 2013 (28.6 age of moon), which is proposed to be the spawning season for this species (*Koeda, Ishihara & Tachihara, 2012*). To observe migration patterns, we collected adult (100–120 mm standard length) *P. schwenkii* from a large school using hand nets or gill nets (with special fish collecting license no. 21–71 from Okinawa Prefecture), and attached a chemical luminescent tag (produced for nighttime angling: LUMICA Co., Ltd., Kemihotaru 25: diameter 2.9 mm, length 23 mm, 0.15 g, duration 3 h, visibility distance 15 m; https://lumica-shop.com/) to the caudal peduncle of the fish by using a cable-tie. We released the tagged fish into the school and monitored it for more than 10 min to confirm that the tagged fish rejoined the school and swam together with other individuals without physical damage and/or aggression towards the glowing tag (Fig. 1). No damage or aggression from other fish was ever observed for the tagged fish in the present study.
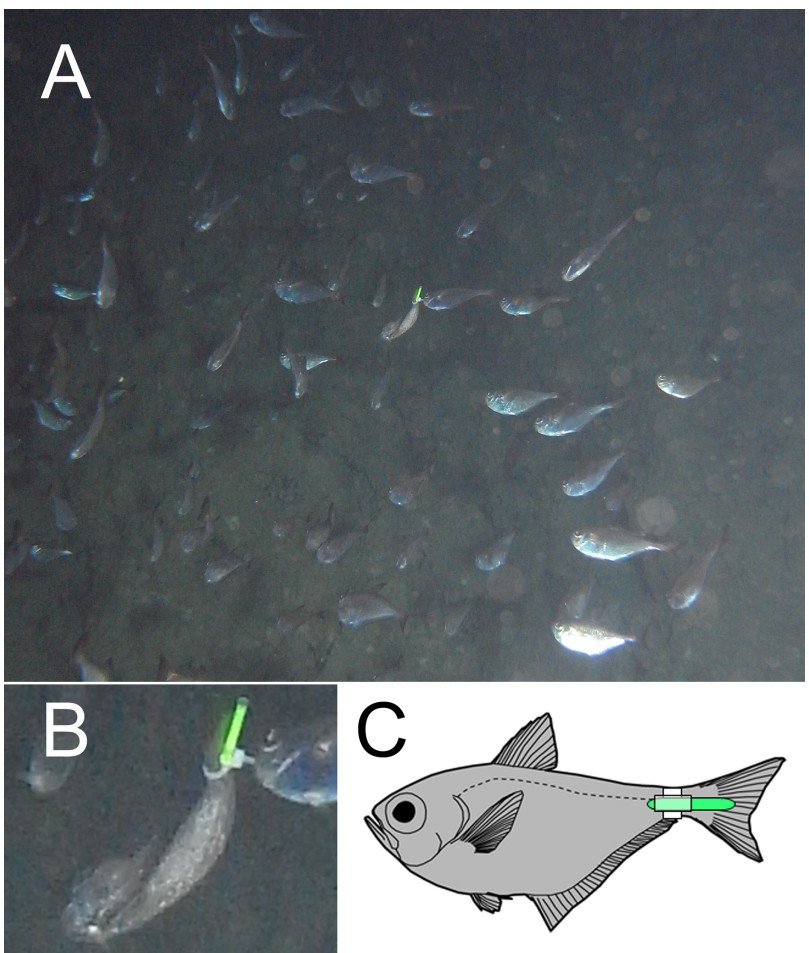

**Figure 1 Photographs and illustration of chemical luminescence tagged *Pempheris schwenkii* swimming in the cave with untagged individuals.** (A) Long shot; (B) close up; (C) diagram.

Observations of the migration and behavioral patterns of the tagged individuals began principally from evening twilight periods, generally from ca. 30 min before sunset to 1 h after they started to migrate from the entrance of cave. The beginning time of the migration was recorded. The tagged fish were followed from the surface using a snorkel and a GPS for recording the lat/long data every 30 s. Torch and SCUBA were not used for the observation to reduce the effect of light, and sounds and bubbles of breathing, respectively. The swimming depth of the tagged fish was estimated every 5–10 min. The accuracy of the visual depth estimation was validated by an observer with a dive computer (SUUNTO Di5) diving to the same depth as the tagged fish.

## Data mapping

The lat/log data recorded while tracing the movement of sweepers were plotted on the underwater topographic map. The marginal errors in the plotted data that arose due to mechanical mistakes were rectified as needed. Swimming direction and distance per 30 s recorded by GPS were used to estimate their general habit.
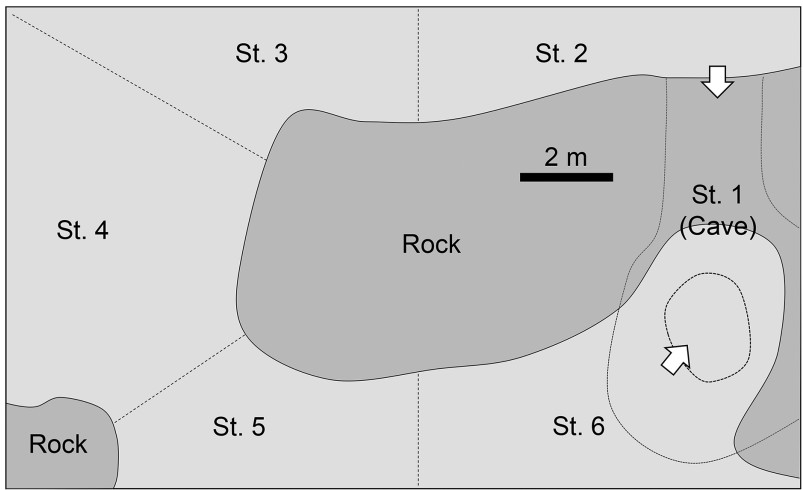

**Figure 2 Diagram of the tank in Churaumi Aquarium viewed directly from above.** Observation area was separated to six stations.

## Aquarium observations

Observations were made in the "Tank of Tropical Sea" in Okinawa Cyuraumi Aquarium (Fig. 2). The volume of the tank is 700 m$^3$ and depths range from 2.5–6.6 m. It is continuously supplied with fresh seawater from the adjacent ocean mixed with circulated filtration seawater, thus the water temperature is almost comparable with that in the field. More than 200 fish species were kept in this tank, including almost 60 individuals of *P. schwenkii*. The tank was subdivided into six areas. The tank contains a large cave (station 1) which has two entrances connecting laterally and longitudinally with stations 2 and 6, respectively. A large artificial rock located at middle of tank. The circulated filtration seawater and fresh seawater supplied to station 4 and 6, respectively. Sweepers usually spent their daytime at cave (station 1) with locating near entrance facing station 2. The number of sweepers in each area and their behaviors were observed every min. The observations were carried out by three to five observers for a total of 3 days, throughout the night, from 18:30 of 30th May 2012 to 6:30 of 31st May 2012, and from 18:30 to 21:00 on 10th June 2012 and 13th June 2012. Light intensity was recorded by a data logger (HOBO pendant logger; Onset Computer Corporation, Bourne, MA, USA) just above the center of the tank. The tank was illuminated by a bright light, which was turned to low-intensity all-night light at 20:00. The light intensity gradually decreased until sunset, and went to almost zero when the main light was turned off.

# RESULTS

## Field observations

During the daytime, *Pempheris schwenkii* usually swam slowly in the open water deep inside the cave and formed one or two large schools. The preparation for nighttime migrations began at 13–35 min (mean: 21 min) after sunset when the entrance of cave was darker (Table 1). Time of sunset ranged between 19:12 and 19:20 during the observation days, but the sky around the cave still gleamed for half an hour. During this

**Table 1 Time schedules and accumulate time of nighttime migration of *Pempheris schwenkii* observed in 5 days.**

|  | 22-May | 27-May | 29-May | 31-May | 8-June |
|---|---|---|---|---|---|
| Time of sunset | 19:12 | 19:15 | 19:16 | 19:17 | 19:20 |
| Start moving to entrance of cave | 19:25 (13) | 19:28 (13) | 19:40 (24) | 19:52 (35) | 19:40 (20) |
| Waiting at outside of entrance | 19:50 (25) | 19:50 (22) | 19:58 (18) | 20:13 (21) | 19:55 (15) |
| Start migration | 20:08 (18) | 19:58 (8) | 20:03 (5) | 20:14 (1) | 20:00 (5) |
| Reach to outer reef | 20:10 (2) | 20:00 (2) | 20:04 (1) | 20:17 (3) | 20:03 (3) |
| Time of observation ended | 20:54 (44) | 21:08 (68) | 21:12 (68) | 21:05 (48) | 21:10 (67) |
| Accumulate time (min) |  |  |  |  |  |
| Total observation time from start migration | 46 | 70 | 69 | 51 | 70 |
| Total time at inner reef | 2 | 2 | 1 | 3 | 3 |
| Total time at outer reef | 44 | 68 | 68 | 48 | 67 |
| Total time at surface water-column | 41 | 21 | 19 | 4 | 68 |
| Total time at shallow water-column | 5 | 30 | 18 | 7 | 2 |
| Total time at deep water-column | 0 | 19 | 32 | 40 | 0 |
| Migrated distance (m) |  |  |  |  |  |
| Total distance of migration | N/A | 430 | 683 | 299 | 920 |
| Calculated total distance of migration per hour | N/A | 379 | 602 | 352 | 786 |

**Note:**
Minutes from previous actions shown in the parentheses.

period, a large school restlessly swam around and moved closer to the entrance. After 15–25 min (mean: 20.2 min), the school moved just outside the entrance and their restlessness reached a peak, during which the fish appeared to be waiting for an individual to lead the migration.

After 1–18 min (mean: 7.4 min), the school suddenly started migrating towards the northeast (60–70°) to the outer reef in small group (Table 1). The time of leaving varied from 19:58–20:14 during the 5 days of observations. The plotted lat/long data of each observed individual (ca. 1 h after they start migrating) are shown in Fig. 3, with total swam distance for each direction. They immediately went through the inner reef and stayed only for 1–3 min (mean ± SE: 2.0 ± 0.45 min, $n = 5$) after they left the entrance of the cave. When they reached the outer reef, the tagged fish mostly swam alone except for a few times when they met and swam together with other sweepers. Their migration patterns in outer reef differed day by day. In general, however, they spent almost all of their time in the area where the depth of sea bottom ranged from 5–30 m, and reached a maximum of 150 m offshore from the reef edge. In the offshore area (outer reef), the observer could not see anything except for a glowing tag, and perfect dark surrounding the fishes. They usually spent time near the surface water-column (less than 5 m depth; white plots in Fig. 3) or shallow water-column (between 5–15 m depth; light gray plots in Fig. 3), but sometimes swam into the deep water-column (more than 15 m depth; dark gray plots in Fig. 3). On 2 days (22nd May and 8th June), the target fishes swam southeast (120–140°) parallel to the reef-edge in the offshore area (Fig. 3). These horizontally swimming individuals spent almost all their time near the surface. For the other 3 days (27th, 29th, 31st May), the target fishes swam vertically around the outside of the reef, and the distance

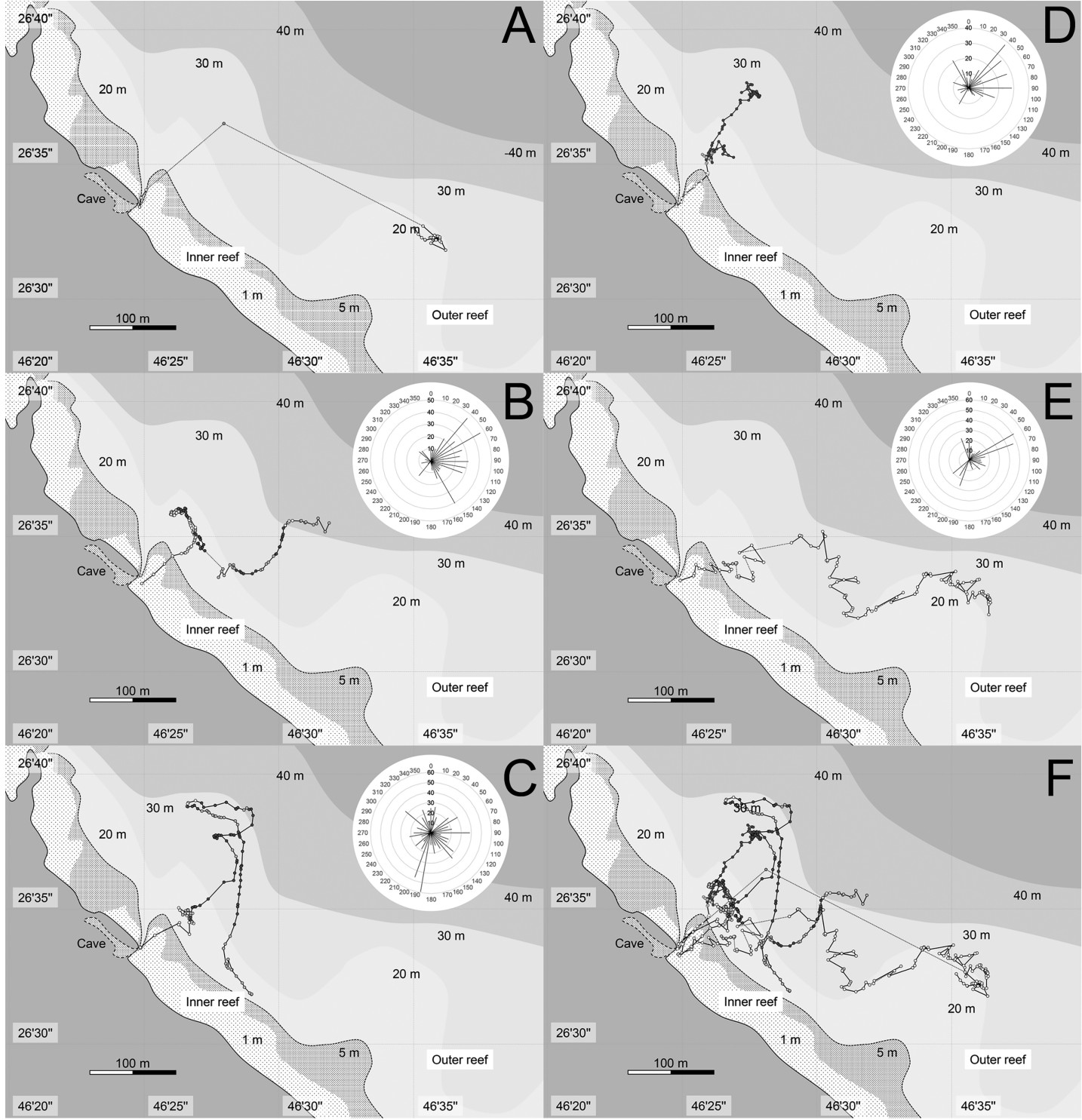

**Figure 3** **Tracks of migration of *Pempheris schwenkii*.** Plots indicating lat/log data for every 30 s of the first an hour after they start migration. The estimated swimming depth in each data are shown by different colors of plots, such as white (surface water-column: <5 m depth), light grey (shallow water-column: 5–15 m depth), and dark grey (deep water-column: >15 m depth). The accumulated swimming distance (m) of each direction is shown at upper right. (A) 22-May; (B) 27-May; (C) 29-May; (D) 31-May; (E) 8-June; (F) 5 days combined.

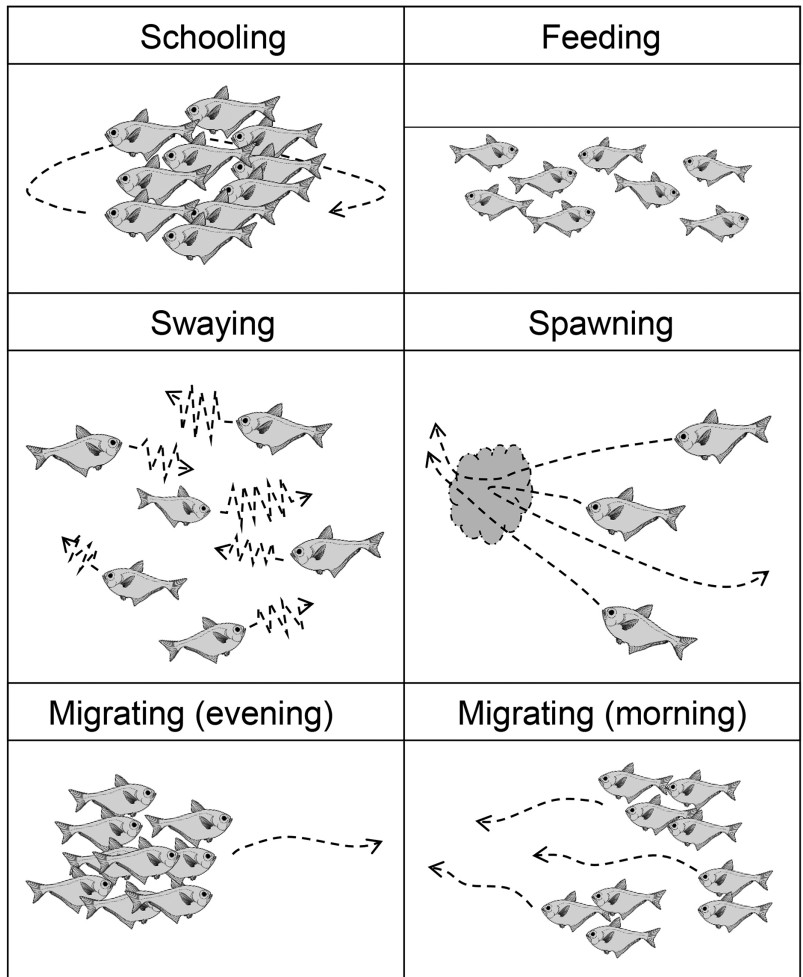

**Figure 4 Illustration of five patterns of behavior of *Pempheris schwenkii* as observed during tank observation.**

of the horizontal migration was shorter than those of the previous 2 days. The accumulated distance of their migration after ca. 1 h from when they started migrating was 379–786 m (mean ± SE: 530.0 ± 102.1, $n = 4$). The longest migration was observed on 8th June, when the sweeper swam straight to the east, and the shortest was observed at 31th May, when it spent most of the time in the deep water-column.

## Aquarium observations

The observed behavior of *P. schwenkii* in the tank was categorized into five main patterns as follows (Fig. 4): schooling in a single crowded school, with individuals usually facing the same direction; shaking and close together but swimming randomly; migrating, purposefully moving a long distance; spawning (oviposition-like behavior); and feeding. The observed time schedule and location of the school are presented in Fig. 5. Sweepers were "schooling" around the cave (station 1) during the daytime. When the time was closer to the natural sunset (19:17–19:22), the light intensity decreased (200–400 lux). During this period, the school went back and forth between outside (station 2) and inside

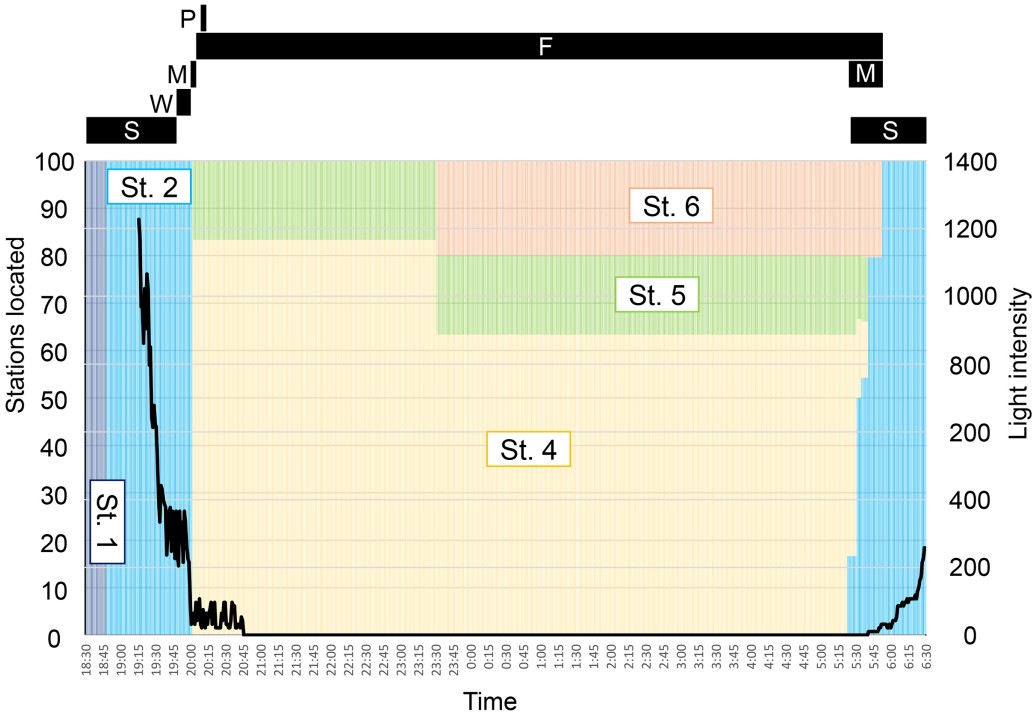

**Figure 5** Time schedule of behavior (black bars) and locality (colored bars) of *Pempheris schwenkii* as observed in tank observation with varying light intensity (black line). The characters indicate observed five behavioral patterns as follows: S, schooling; W, swinging; M, migrating; F, feeding; P, spawning.

(station 1) of the cave. Approximately 5 min before the daytime light turned completely off at 20:00, all of the "schooling" fish suddenly started "shaking". This behavior was observed on 30th May, but not on 10th and 13th June. After a few min of this "shaking", when the daytime light turned off, all fishes immediately started "migrating" from station 2 to the corner of station 4. They waited for several minutes after arriving at station 4, and "spawning" was observed at 1 m below the surface. A single individual, inferred to be a female, turned and swam rapidly, and released a cloud of egg-like material into the water. Two fish, assumed to be males, immediately swam into the cloud, but release of sperm was not observed. This spawning activity was observed only on 30th May, and not on the other two observation days. After the "spawning", the fish dispersed to stations 4–6 and started "feeding" near the surface throughout the night. In the morning, 10 fish started "migrating" to the cave (station 1) at 5:23, and 22 fish followed them at 5:31 just before the light intensity started to increase from 5:40 (ca. 10 lux). Approximately 10 min later (5:53; ca. 30 lux), the remaining fish at stations 4–6 swam back to the cave in small groups, and started "schooling" in the cave.

## DISCUSSION

To the best of our knowledge, this was the first attempt to follow nocturnal fish under darkness using chemical luminescent tags. In this method, we could clearly observe the small-scale but dynamic nocturnal migration pattern of *P. schwenkii* in nature, which is

difficult to observe using other methods. In addition, combining the results of our nature and aquarium observations, the behavior of *P. schwenkii* can be divided into the following processes: (1) school in caves during daytime; (2) move to the entrance and wait for migration; (3) share a sign (shaking; spawning season only) when they spawn in the night; (4) quickly migrate from inner reef shelter to offshore outer reef; (5) spawn at outer reef (spawning season only); (6) disperse and feed on zooplankton in offshore open water-column for several kilometers; (7) return to their diurnal shelter before the day becomes bright.

During the observation in the aquarium, spawning behavior was exhibited for a few minutes after the migration from cave to the farthest station (station 4). This fact indicating that they are avoiding to spawn near their daytime shelter. In addition, the nocturnal migration observed in nature indicated that they immediately migrate to outer reef after they start migration. From these results, it can be inferred that they are likely to spawn after migrating to the outer reef. Spawning in the outer reef is well known in other fish groups, and reduces the risk of egg predation (*Johannes, 1978*). The nighttime spawning of *P. schwenkii* is also supported by the gonadal data, because evening (15:00 to sunset)-collected specimens have the most developed ovaries in their gonads, and morning (sunrise to noon)-collected specimens most frequently exhibit postovulatory follicles, indicating they have recently spawned (*Koeda, Ishihara & Tachihara, 2012*). *Johannes (1978)* indicated that nighttime spawning decreased the predation risk for not only eggs by the plankton feeder but also of adult fishes by piscivorous organisms. This indicates the specific adaptation of nocturnal fishes. Finally, *P. schwenkii* may reduce their predation risk by exhibiting two behaviors, which are spawning at outer reef and at nighttime.

In the aquarium, shaking behavior by all captive individuals was observed only on the day that spawning occurred. This behavior may be characteristic of spawning; however, more trials are warranted in this regard in the future. In spite of 60 individuals being kept in the aquarium and observations being carried out during their spawning season, the spawning behavior was observed only once in 3 days. This observation may be attributed to several possible reasons: they are too old for spawning (more than 8 years old), whereas the maximum age has been estimated to be 6 years old, and most of the individuals younger than 2 years old in nature (*Koeda et al., 2016*); the spawning season ended in the tank, since the spawning season is estimated to be January to June (*Koeda, Ishihara & Tachihara, 2012*) and observations were made in late May and June; or spawning was suppressed due to the stress of the aquarium environment.

*Annese & Kingsford (2005)* and *Fishelson, Popper & Gunderman (1971)* tagged and released sweepers in daytime shelters and observed them continuously at the same sites, revealing that most of the sweepers returned to the same shelter over the following 7 weeks to 3 months. The results of the present study supported the notion that all the fishes in the aquarium observation came back to the cave (station 1). In addition, the fishes tagged in our preliminary research carried out on 18th May were observed again at 22nd May in the same cave.

Assuming they keep moving at the same speed and active nighttime is 10 h/day during the observed period, it was estimated that they can potentially migrate several kilometers

(up to ca. 7 km, minimum 4 km) in one night. Even they if they immediately came back from the point we finished the observation to their home cave, it means they can potentially migrate at least 2 km in one night. It is well known that reef fishes are visually familiar with their surroundings, and often use landmarks during migrations (*Bardach, 1958*; *Hobson, 1968*, *1974*; *Ogden & Buckman, 1973*). *Gladfelter (1979)* also agreed with this notion based on their observation of the migration of *P. schomburgki*. However, the present observation indicated that *P. schwenkii* usually swam in the shallow water column in the 20–30 m depth area, which is sometimes more than 100 m offshore from the reef edge. It may indicate that the darkness as perceived by human vision is still light enough for sweepers to recognize their position from the topographic landscape.

On a coral reef many big fishes start to hunt at twilight. Most diurnal fishes fall asleep in this time zone, and are vulnerable to piscivores. *Hobson (1968*, *1972*, *1974*) found that there is a quiet period of 15–20 min duration between the disappearance of both diurnal and nocturnal fishes (in the evening or morning, respectively) and the emergence of nocturnal and diurnal fishes. Considering this fact, *P. schwenkii* wait at the entrance of cave for almost half an hour after the sunset, which indicates an attempt to reduce the predation risk from diurnal piscivorous fishes during this period. Therefore, they may start migration all at once after waiting for a while, which is deemed to have reduced the activities of nocturnal piscivorous fishes, and swim out to the offshore outer reef immediately. In our study, *P. schwenkii* swam near the surface or in the shallow water-column during the nighttime and were far away from the bottom or walls where there are structures that can shelter them from predators. In addition, they have no agility to escape from large piscivorous fishes, and neither do they have strong spines or scales to avoid predation. It is easy to imagine that they are preyed upon by large piscivorous fishes if they are found in the open water column at outer reef. In fact, members of the family Pempheridae are recorded from the stomachs of various large piscivorous fishes, such as Sphyrnidae, Serranidae, Carangidae, Lutjanidae, and Trichiuridae (*Hobson, 1968*; *Shpigel & Fishelson, 1989*; *Koeda & Motomura, 2017*). It indicates that the sweepers are important food resources for diversified piscivorous fishes that prey during the nighttime. Nonetheless, it can be said that the density of large piscivorous fishes in the outer reef open water column should be comparatively lower than that of fishes in the inner reef and reef edge. In exchange for the risk of predation, they can feed on the rich nocturnal zooplankton (including meroplankton) in the open water column. These zooplankton assemblages consist of diverse invertebrates, especially small, motile crustaceans, which are more abundant in exposed positions during night than during the day (*Longley, 1927*; *Hobson, 1965*, *1968*; *Starck & Davis, 1966*). The composition of the available plankton differs markedly between day and night, with generally larger meroplanktonic forms (especially crustaceans and annelids) rising from the bottom into the water column after dark (*Alldredge & King, 1977*; *Porter & Porter, 1977*; *Esquivel-Garrote & Morales-Ramírez, 2020*). Only a few swimming fish groups, such as Clupeidae and Atherinidae, may compete for this rich food resource with Pempheridae

(*Hobson, 1974*). *Koeda et al. (2016)* compared the age and growth of *P. schwenkii* and *P. adusta* in the Okinawa Island, and indicated that the growth coefficient (K) of *P. schwenkii* was significantly higher than those of other reef fishes and that the species can grow up to more than 80% of their maximum length in the first year. In addition, *Koeda, Ishihara & Tachihara (2012)* indicated the spawning of *P. schwenkii* was not related to the lunar cycle, and occurred more frequently compared to other reef fishes. Their rapid growth and frequently reproduction may be attributed to their feeding behavior in that they can full up their stomach every night with rich zooplankton from outer reefs. The first author opened the fully filled stomachs of *P. schwenkii* collected early morning at the same cape, and found them filled with zooplanktons, such as Crustacea (megalops, copepods, and marine water striders), Polychaeta, fish eggs, and larvae. *Pempheris adusta* was observed in the inner reef area during nighttime in the present observations and not at outer reefs. Previously, *Koeda et al. (2016)* showed that the growth coefficient of *P. adusta* was almost half of *P. schwenkii*, which might be a result of the difference in their feeding habits.

The observations in the nature and aquarium in the present study, combined with previous studies, clearly showed that sweepers inhabited in inner reef during daytime and migrated to an offshore outer reef to feed on rich zooplankton. This finding could be important in evaluating their role in the coral reef ecosystems. Firstly, they are considered to be an important prey item for the piscivorous fishes, which is usually located on the top of the coral reef ecosystem. Their growth rate is higher than many other groups (*Koeda et al., 2016*), and they provide stable resources for the predator fishes. Secondly, they prey on rich zooplankton in the outer reef, import them to their daytime shelter on the inner reef, and evacuate them in their feces. A large amount of feces from a great number of sweepers may act as a nutrition source that supports the dark environment, such as caves and crevasses, which are oligotrophic areas in the coral-reef ecosystem. In other words, sweepers are not only a food resource for large carnivorous fish in the coral reef area but also play a role in transporting nutrients and energy from the rich outer reef into poor inner reef. These characteristic migration patterns have not been reported for other nocturnal zooplanktivorous fishes, such as Holocentridae and Apogonidae, and most of them swim near their daytime shelter (K. Koeda & K. Tachihara, 2021, unpublished data). Although some other migrating fishes, such as Clupeidae and Atherinidae, also share rich food resources in outer reef during the nighttime, they are nomadic and do not stay in a single coral reef. Therefore, the characteristic behavior of Pempheridae observed in the present study may be unique to this family. A similar importation pattern of organic carbon was reported for the blacksmith (*Chromis punctipinnis*) which is a diurnal zooplanktivore in southern California waters (*Bray, Miller & Geesey, 1981*). However as mentioned above, the richness of zooplankton in the water column is dramatically increased in the nighttime (*e.g.*, *Alldredge & King, 1977*), and the long-distance migration of the Pempheridae observed in the present study should have greater importance compared to that of blacksmith. Finally, the unique behavior of Pempheridae indicates a possibility of them making an important contribution to flows of energy and materials in coral reef ecosystems.

# ACKNOWLEDGEMENTS

We are especially grateful to S. Koizumi (previously University of the Ryukyus) for his dedicated support in field and tank observations, and giving useful advice on the research planning. We thank H. Miyahara, M. Nonaka, S. Matsuzaki (Okinawa Churaumi Aquarium), T. Fujii, H. Saimaru, H. Suematsu, I. Kawamura, T. Kunishima, Y. Fujiwara, K. Araki, M. Sakurai, H. Kise and K. Sakai (previously University of the Ryukyus) for their help in field and/or tank observations. We also thank F. Ziadi-Künzli (Okinawa Institute of Science and Technology) for giving useful advice on the manuscript preparation.
The English in this document has been checked by at least two professional editors, both native speakers of English (EigoExperts Pvt. Ltd.).

## Funding

Keita Koeda was supported by a Grant-in-Aid from the Japan Society for the Promotion of Science for JSPS Fellows (DC1: 23-2553; PD: 26-477), and JSPS Overseas Research Fellowships (29-304), JSPS KAKENHI 21K06313 JP, and the Sasakawa Scientific Research Grant from The Japan Science Society. The funders had no role in study design, data collection and analysis, decision to publish, or preparation of the manuscript.

## Grant Disclosures

The following grant information was disclosed by the authors:
Japan Society for the Promotion of Science for JSPS Fellows: DC1: 23-2553; PD: 26-477.
JSPS Overseas Research Fellowships: 29-304.
JSPS KAKENHI: 21K06313 JP.
The Japan Science Society.

## Competing Interests

Hideyuki Touma is employed by the Okinawa Churaumi Aquarium.

## Author Contributions

- Keita Koeda conceived and designed the experiments, performed the experiments, analyzed the data, prepared figures and/or tables, authored or reviewed drafts of the paper, and approved the final draft.
- Hideyuki Touma performed the experiments, authored or reviewed drafts of the paper, and approved the final draft.
- Katsunori Tachihara conceived and designed the experiments, authored or reviewed drafts of the paper, and approved the final draft.

## Animal Ethics

The following information was supplied relating to ethical approvals (*i.e.*, approving body and any reference numbers):

We do not have an approval system in Japan for research.

### Field Study Permissions

The following information was supplied relating to field study approvals (*i.e.*, approving body and any reference numbers):

Special fish collecting license from Okinawa Prefecture (no. 21–71).

### Data Availability

The observation data (lat/long with time schedule) of *Pempheris schwenkii* is available in the Supplemental File.

### Supplemental Information

Supplemental information for this article can be found online at http://dx.doi.org/10.7717/peerj.12412#supplemental-information.

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
