# Peer review of "Nighttime migrations and behavioral patterns of Pempheris schwenkii"

_PeerJ, doi:10.7717/peerj.12412_

## Round 0.1 · original submission · Major Revisions

I found your manuscript interesting. The two reviewers disagreed as to its suitability for publication but I am happy to consider a revised version if you wish to resubmit. In addition to the reviewers' comments, could you please also address the following points:

The conclusion raises a new point about the potential for the fish to import organic matter into their cave in the form of faeces. This should be mentioned in the main part of the Discussion (not the conclusion). This article is relevant:
Bray, R. N., Miller, A. C., Geesey, G. G. 1981. The fish connection: a trophic link between planktonic and rocky reef communities? Science 214: 204-205.
Replace 'despise' with 'aggression towards" and 'despising' with 'aggression'.

Reviewer 1 ·

Basic reporting

Re: Manuscript 62502

General comments:
The paper aligns with the intent of PeerJ

The subject of the paper is of interest and the type of data presented on nocturnal fish are rare.

The combination of field and tank experiments was sensible. However, substantial revision is required.

In general, there was poor utilisation of the data available. It was clear that data were collected and mentioned somewhat casually in the text of the results, but we’re not presented in figures or tables. I have made a number of suggestions.

Some editorial work required, especially with respect to word usage.

In general, the captions to the figures were incomplete.

Specific comments
Use past tense for material and methods and results.

Materials and methods
Visibility of tag (by distance?) and duration the tag held its glow? State how the tag was attached to the peduncle?
Line 105 and 138 – re mapping – what about landmarks such as large rocks and like?
Line 146 re: 200 fish, 60 Pempheris – what were the rest?

Results
Line 157 before starting the field.
Line 171 – 172 – fish spent about 2 minute in in-reef then to out reef, but n lines 178-180 – they usually spent time in shallow water – clarify!
Line 172 re: (Mean 2.0) give SE and sample size
Line 174, give percentage of time as an individual and as a group?
Line 176, Seriously without SCUBA how mapping to 30m and even more unlikely tracking to 150 m achieved?
Line 178 - they usually spent time in shallow water? This is unclear from the maps. If lots of individuals tracks provide time versus depth budgets in a table with all replicates
Lines 181- 189 - variation in trackers with time not presented as table # of graphs.

Figures
Fig 1 – provide an estimate of size of fish and the size of the tag. The tags are attached with a cable tie?
Fig 2 needs a scale
Fig 3 caption - specify for individuals A-D. There is a need for a table that includes all individuals – e.g., time spent at inner and outer areas of reefs and rate of movement/hr or minute.
No need to present depths as minus values
Fig 4 sexes of fish in the spawning image?
Figure 5 Feeding behaviour may be best presented as a line - from essentially sunset to sunrise? In the caption Specify the number of nights that this plot was developed from and the number of fish.
So site 3 and 2 were not used!?

Discussion
Line 216 “chase” = follow?
Lines 219-223 – these are results!
Line 229 - 238 inference that may spawn at outer reefs - delete as no evidence for this.
Lines 238 to 248, comment - spawning may not occur because stressed in aquarium, or not the right environmental triggers. Sampling frequency did not allow for the possibility that spawning only happened on so nights (e.g., semilunar).
Line 249 Annese & Kingsford found quite different behaviour by species. One species was found in the shallows in groups, while the other was rather solitary and stayed in deeper waters
Line 255 - Re migration several kilometres - little evidence of this. An hourly rate of movement was calculated, but chances are they spent most of the night foraging once the reached a targeted area?
Lines 307 - 326, any conclusions are made that do not relate to the results. In fact, little is said here on the nature of migrations - which is what the paper is about!
Examples of errors Wrong use of words:
Line 25 – ‘nocturnal fish group’ should be ‘nocturnal group of fish’
Line 35 – re: “on surface” you mean near the surface? Correct throughout.
Line 109 and elsewhere. “In-reef” and “out-reef” would be better as inner reef and outer reef, change throughout
Line 126 ‘despising’; line 128 were not was; line 147 ‘comparted’;
Line 161 re: gleamed” you mean there was still ambient light?
Line 164, waiting for and individual to lead the migration; line 301 lowers caps for polychaeta.
In-reef and out-reef = inner and outer reefs.

Experimental design

all comments are in the first box

Validity of the findings

all comments are in the first box

Additional comments

all comments are in the first box

Reviewer 2 ·

Basic reporting

Many lapses in English and poor choice of wording. It really needs another thorough revision to improve the English and choice of words. "Chasing" is used inappropriately. This should be changed to "Following" as we hope that the diver observing fish movements is not actually chasing the fish, which surely would affect its movements. In therms of the behavioural category "swaying"... this is not a term that is currently in use. Is there an accepted behavioural term that could be substituted here.

The literature is largely out of date and not used to generate any hypotheses.

Experimental design

Research questions were not clearly defined.The basic sampling design is really far to minimal to make quantitative assessment possible. This is really just a collection of anecdotal observations.

Validity of the findings

The use of the luminescent tags is probably novel. But any likely effects of the tags on behaviour were not assessed. The sample size was too low and I do not think it is appraise to publish "rough" back of the envelope estimates of movement distances. This can only be improved by further work.

Annotated reviews are not available for download in order to protect the identity of reviewers who chose to remain anonymous.

---

## Round 0.2 · Minor Revisions

Thank you for addressing the reviewers' comments in your revision. I have made a number of minor edits to the attached MS (using track-changes) and asked some questions in the comments. Please check that you approve of the edits, and make any further changes in relation to the comments.

---

## Round 0.3 · accepted · Accept

Thank you for making the requested revisions.